# Twisted Gastric Tube after Laparoscopic Sleeve Gastrectomy—An Unusual but Effective Surgical Approach to Achieve Full Recovery

**DOI:** 10.3390/jcm11092304

**Published:** 2022-04-20

**Authors:** Gerardo Sarno, Pietro Calabrese, Salvatore Tramontano, Luigi Schiavo, Vincenzo Pilone

**Affiliations:** 1“San Giovanni di Dio e Ruggi D’Aragona” University Hospital, Scuola Medica Salernitana, Via San Leonardo, 84125 Salerno, Italy; 2Center of Excellence of Bariatric Surgery of the Italian Society of Obesity Surgery and Metabolic Disease (SICOB), Unit of General and Emergency Surgery, University Hospital San Giovanni di Dio e Ruggid’Aragona, P.O. Gaetano FucitoMercato San Severino, 84125 Salerno, Italy; pietrocalabres@gmail.com (P.C.); salvytra@libero.it (S.T.); lschiavo@unisa.it (L.S.); vpilone@unisa.it (V.P.); 3Department of Medicine, Surgery and Dentistry, “Scuola Medica Salernitana”, University of Salerno, 84081 Baronissi, Italy

**Keywords:** sleeve gastrectomy, gastric twist, surgical complications, gastric bypass, bariatric surgery, obesity

## Abstract

Sleeve gastrectomy is at present the most practiced bariatric intervention for patients suffering from severe obesity. Although rare, post-operative complications such as leakages and strictures may represent a challenging issue for bariatric surgeons and cause impaired quality of life for patients. Gastric twist is even more rare. This complication is a functional obstruction rather than a stricture of the gastric remnant most likely due to technical mistakes at index surgery. If endoscopy usually allows diagnosis and constitutes the first-line treatment for this condition, surgery is mandatory when endoscopy is not successful. The conversion of the sleeve to a Roux-en-Y gastric bypass is the usually chosen intervention but a wide range of reconstruction has been proposed. In this report, we discuss the surgical technique we employed to achieve a full resolution of a gastric twist.

## 1. Introduction

Bariatric surgery is the most effective intervention for sustained weight loss and long-term resolution of weight-related comorbidities in patients with severe obesity [1,2]. Sleeve gastrectomy (SG) is at present the most practiced surgical approach due to its safety and effectiveness [3,4,5].

Complications after SG (mainly leakages and strictures) are rare, but may represent a challenging issue for bariatric surgeons and cause impaired quality of life for patients [6]. Strictures have to be considered in case of worsening gastroesophageal reflux disease (GERD) despite medical therapy, persistent nausea and repeated vomiting [6,7,8]. Diagnosis is obtained with upper gastrointestinal contrast imaging (UGI), while endoscopy allows diagnosis of functional strictures due to gastric helical twist [5,9] and constitutes the first-line treatment for strictures with a reported success rate of up to 95% [9]. When endoscopy fails, revision surgery is mandatory [5,9]. At present there is not a definitive consensus on the surgical procedure to perform. Although converting the sleeve to a Roux-en-Y gastric bypass (RYGB) is usually chosen [10] a wide range of reconstruction have been proposed [4,11].

Hereby we discuss our approach in a patient referred to our Unit for a gastric twist (GT) following SG.

## 2. Case Presentation

A 43-year-old male patient with severe obesity was referred to our unit six months after SG since our institution is a tertiary referral center for bariatric surgery and for the management of bariatric complications. The patient, whose weight before SG was 130 kg (BMI: 44), on admission suffered worsening GERD, despite sustained therapy with PPI (40 mg/day pantoprazol), aroused at post-operative day 3 after index intervention, nausea, repeated non-bilious vomiting, epigastric pain and fullness. No other remarkable comorbidities were detected. His weight on admission was 89 kg (BMI: 30).

UGI revealed a hold-up of the contrast in the distal esophagus and upper part of the sleeve, with a delayed gastric emptying suggestive for GT (Figure 1).

Endoscopy confirmed the diagnosis, but an attempt to place a fully covered stent was unsuccessful, because of a stent displacement noted the day after the procedure. Surgical revision by laparoscopy was planned. Pneumoperitoneum (12 mm Hg) was obtained through a Verres needle in the left hypochondrium, and five trocars were placed. Lysis of hepatic, omental and mesocolic adhesions up to reach the angle of His allowed an adequate exposure and a careful inspection of the stomach. The twist involved the upper third of the sleeve because a bypass was chosen. Jejunum was divided 50 cm below the ligament of Treitz to create the bilio-pancreatic limb. Subsequently, a side-to-side gastrojejunostomy above the sleeve stenosis was created in an antegastric and antecolic fashion by using a laparoscopic linear stapling device (60 mm blucartdrige) with suture closure of the defect. The stomach was not divided to preserve the gastro-duodenal transit. The biliopancreatic limb was anastomosed to the distal segment of jejunum (75 cm from the jejunal division) to create side-to-side jejunojejunostomy with a laparoscopic linear stapling device (60 mm white cartridge) with suture closure of the defect. The methylene blue test was negative. Mesenteric defects were sutured, and a surgical drain was placed close to the anastomosis. There was no blood loss.

The post-operative course was uneventful and the patient fully recovered from obstructive symptoms. Post-operative day one UGI showed a normal progression of the contrast agent both into the duodenum and through the gastric bypass (Figure 2).

The patient was discharged at post-operative day 3. One month after surgery, GERD symptoms were well controlled with 15 mg/die lansoprazole. At six months follow-up, the patient’s weight was 73 kg (BMI: 26), without symptoms of GERD and PPI discontinued, with self-reported excellent quality of life.

## 3. Discussion

Herein, we describe the surgical approach in a patient suffering obstructive symptoms following SG, who successfully recovered with a gastro-jejunal bypass performed preserving the gastro-duodenal transit. Complications of SG are rare and account for about 2% of the procedures performed [12]. Gastric twist is a well-recognized complication of SG, due to a functional stenosis caused by an unequal traction on the anterior and posterior wall of the stomach and subsequent spiral stapling at surgery [9]. The onset of obstructive symptoms may appear early in the post-operative course [13], but also late presentation with worsening of GERD, epigastric pain and dysphagia have been reported [14]. Patients with obesity are prone to GERD for several reasons such as weak lower esophageal sphincter and altered gastroesophageal junction pressure gradient [15,16]. SG is the bariatric procedure with the higher risk of GERD development because of the increase in intragastric pressure, along with a slow food transit time through the pylorus [17]. Bypass following SG is the treatment of choice for post-sleeve gastrectomy GERD [10,16,18,19].

The incidence of gastric twist as a complication of SG is still not perfectly clarified, most probably because it is misdiagnosed and so underestimated. In the recent literature, the incidence of stenosis following SG ranges between 0.69% and 2% [20,21]. In the manuscript by Abd Ellatif 2017 et al. [9], the detected incidence of GT was 1.23%, while Redibo et al. [5] reported a total incidence of gastric stenosis of 1.4%; GT was diagnosed in 41.2% of cases.

Endoscopy by balloon dilatation and/or stent placement usually warrants a full recovery and relief of symptoms, with revisional surgery employed after failure of any other attempt [4,5,9], or as first-line treatment if expressly requested by the patient [4]. Revision surgery usually foresees a conventional RYGB with stomach transection [8], since resection of the affected portion of the stomach is thought fundamental [4]. Gastric division usually leads to the derotation of the two parts of the stomach. This evidence indicates the technical error occurred at the initial stapling [6,7]. In our opinion, in the absence of specific reasons (e.g., vascular compromising of the gastric walls, underlying fistula, etc.), the resection of the stomach should be avoided. As a matter of fact, by dividing the stomach, endoscopic evaluation of both the upper gastrointestinal and the biliary tract is no longer possible. The impossibility to guarantee this evaluation in those patients with a higher risk for cancer or biliary diseases is a major issue. Preserving the stomach and the duodenal chymus pathway reduces the risk of iron and vitamin B12 deficiency. Furthermore, gastrointestinal hormone secretion, physiological biliary and pancreatic enzyme outflow are maintained [22]. Finally, the adopted approach avoids blind loops and narrow anastomoses, can be easily reversed or converted and reduces the risks of dumping and diarrhea.

Strategies to preserve the endoscopic access to the gastric remnant and duodenum were already proposed. Oshiro et al. [4] described a reconstruction with three anastomoses following resection of the affected part of the stomach: esophagojejunostomy, gastrojejunostomy created 15 cm below the previous anastomosis with a stoma 10 mm in width to allow the passage of a standard endoscope, and jejunojejunostomy. In our opinion, such an approach, although effective, is not as easy as the reconstructive strategy we adopted. As a matter of fact, the need for gastric resection and the subsequent need to perform three anastomoses increases the complexity of the intervention as well as the risk of post-operative complications such as hemorrhages, leaks and infections. Additionally, the development of strictures over the follow-up might result in the impossibility to perform endoscopy and so in a failure of the treatment proposed.

The operative risk of revisional bariatric surgery increases when compared to primary procedures. In this view, careful patient evaluation and meticulous surgical technique are mandatory. Surgery has to be minimally invasive and effective, aiming to reduce the risk of malabsorption and to minimize irreversible remodeling of the gastrointestinal tract.

## 4. Conclusions

Bariatric surgery and the surgical treatment of bariatric complications are a fascinating field of surgery with the success of intervention strongly related to the experience and technical skills of bariatric surgeons. The described approach succeeded in the management of GT. If validated in a large cohort study, it may represent in future an alternative choice when non-surgical treatments fail to give a definitive solution.

## Figures and Tables

**Figure 1 jcm-11-02304-f001:**
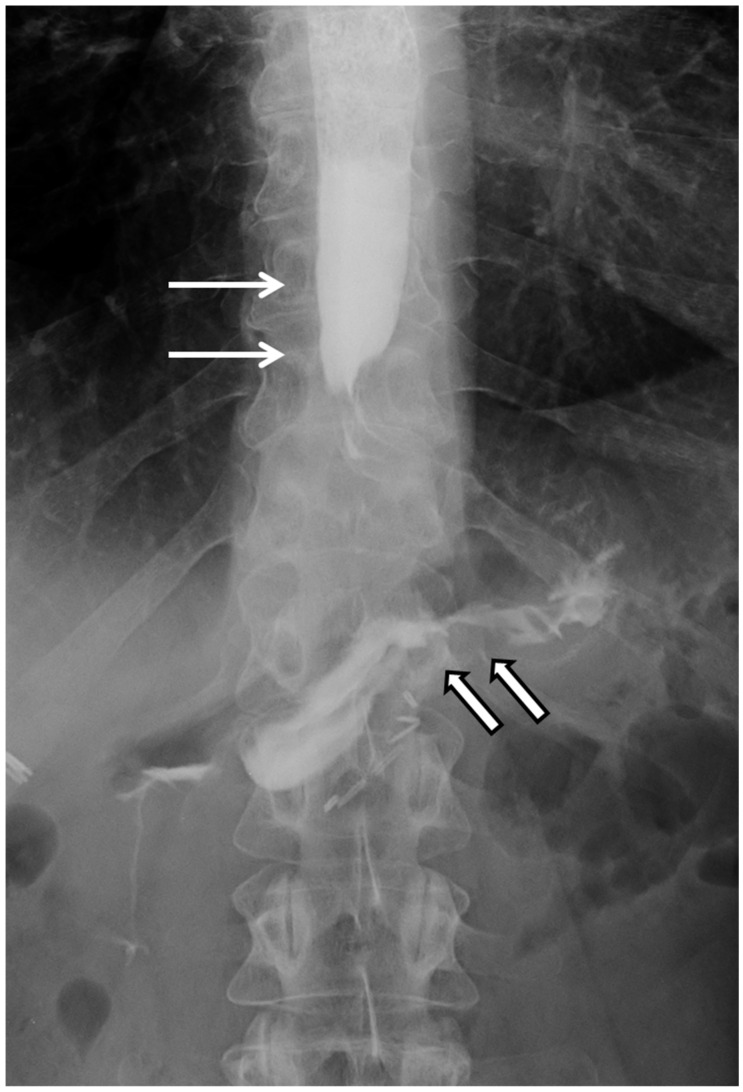
Post-sleeve gastrectomy upper gastrointestinal contrast. Hold-up of the contrast in the distal esophagus and upper part of the sleeve (thin arrows), with delayed gastric emptying sustained by gastric twist above the level of the incisura angularis (thick arrow).

**Figure 2 jcm-11-02304-f002:**
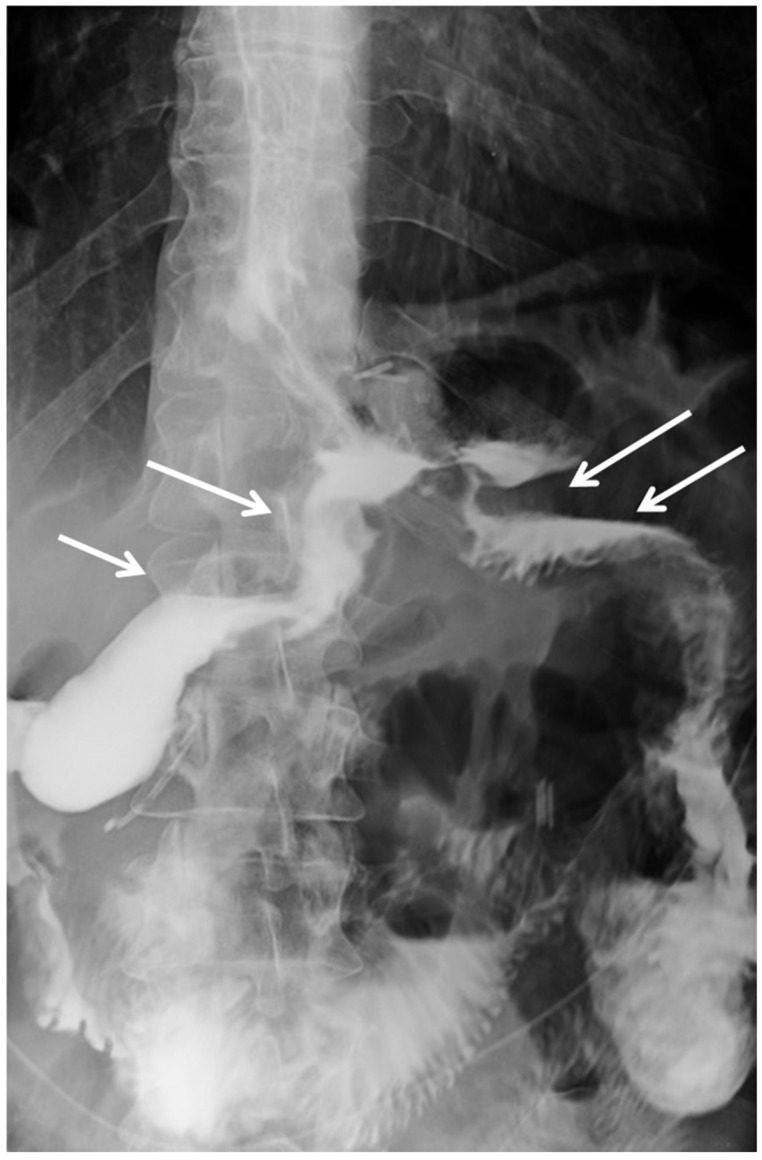
Post-operative upper gastrointestinal contrast. Normal contrast flow into duodenum (arrows) and through the gastro-jejunal bypass (arrows), without any stay of contrast material in the remnant stomach.

## Data Availability

Not applicable.

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
