# Peer review of "Twisted Gastric Tube after Laparoscopic Sleeve Gastrectomy—An Unusual but Effective Surgical Approach to Achieve Full Recovery"

_jcm, 2022, doi:10.3390/jcm11092304_

Round 1

Reviewer 1 Report

The authors presents a nice case of gastrointenstinal twist post SG and their surgical approach to solve the issue. I would recommend the following to strengthen the case. 

Major: Could the authors mention the patient weight before his bariatric surgery, after the surgery but before their gastrojejunostomy, and the patient weight 6 months after their procedure. This would be important for readers to know since weight loss was the main indication for his surgery in the first place.

Minor:

  • There are minor grammatical errors throughout the manuscript. For example: Line 50: “suffered of worsening” should be “suffered worsening”. Line 51 and 84: 40 mg/die, should that be mg/day?
  • Line 51: aroused early, could the authors give a specific time line of when this happened (i.e. 5 days, 10 days etc). Early is subjective.
  • Line 64, angle of his should be angle of “His”
  • Line 87: suffering of obstructiveà suffering obstructive
  • Line 99 there is a comma in the number where a decimal should be

Author Response

Major:

A: Could the authors mention the patient weight before his bariatric surgery, after the surgery but before their gastrojejunostomy, and the patient weight 6 months after their procedure. This would be important for readers to know since weight loss was the main indication for his surgery in the first place.

R: patient’s weight at index surgery, at referral in our Unit and after six months follow-up has been reported within the text.

Minor:

A: There are minor grammatical errors throughout the manuscript. For example: Line 50: “suffered of worsening” should be “suffered worsening”. Line 51 and 84: 40 mg/die, should that be mg/day?

R: corrected in mg/day

A: Line 51: aroused early, could the authors give a specific time line of when this happened (i.e. 5 days, 10 days etc). Early is subjective.

R: time line - post-operative day 3 - has been specified within the text

A: Line 64, angle of his should be angle of “His”

R: corrected in angle of “His"

A: Line 87: suffering of obstructiveà 

R: corrected in suffering obstructive

A: Line 99 there is a comma in the number where a decimal should be

R: corrected

Reviewer 2 Report

In this case report of a patient experiencing a gastric twist following sleeve gastrectomy, the authors describe the assessment, diagnosis, and surgical treatment. This manuscript contributes to the current literature in which there is not a treatment consensus.

Expand on lines 100-101.

Grammar mistakes make reading the manuscript tricky at points (lines 62-65, 106-108, 110-112).

Much of what was discussed in the case presentation has been published before, and there does not appear to be novelty in this case. However, the discussion offered opinions on various techniques and interventions that would be of greater interest to the reader and should be expanded upon (lines 108+ and 122+).

Author Response

A: Expand on lines 100-101.

R: A revision has been made

A: Grammar mistakes make reading the manuscript tricky at points (lines 62-65, 106-108, 110-112).

R: Grammar revision has been carried out in order to make the text clear.

A: Much of what was discussed in the case presentation has been published before, and there does not appear to be novelty in this case. However, the discussion offered opinions on various techniques and interventions that would be of greater interest to the reader and should be expanded upon (lines 108+ and 122+).

R: A revision has been made
